# Draft-and-Revise: Effective Image Generation with Contextual RQ-Transformer

**Doyup Lee**[*]
Kakao Brain
doyup.lee@kakaobrain.com

**Chiheon Kim**[*]
Kakao Brain
chiheon.kim@kakaobrain.com

**Saehoon Kim**
Kakao Brain
shkim@kakaobrain.com

**Minsu Cho**
POSTECH
mscho@postech.ac.kr

**Wook-Shin Han**[†]
POSTECH
wshan@dblab.postech.ac.kr

## Abstract

Although autoregressive models have achieved promising results on image generation, their unidirectional generation process prevents the resultant images from fully reflecting global contexts. To address the issue, we propose an effective image generation framework of *Draft-and-Revise* with *Contextual RQ-transformer* to consider global contexts during the generation process. As a generalized VQ-VAE, RQ-VAE first represents a high-resolution image as a sequence of discrete code stacks. After code stacks in the sequence are randomly masked, Contextual RQ-Transformer is trained to infill the masked code stacks based on the unmasked contexts of the image. Then, we propose the two-phase decoding, Draft-and-Revise, for Contextual RQ-Transformer to generate an image, while fully exploiting the global contexts of the image during the generation process. Specifically. in the *draft* phase, our model first focuses on generating diverse images despite rather low quality. Then, in the *revise* phase, the model iteratively improves the quality of images, while preserving the global contexts of generated images. In experiments, our method achieves state-of-the-art results on conditional image generation. We also validate that the Draft-and-Revise decoding can achieve high performance by effectively controlling the quality-diversity trade-off in image generation.

## 1 Introduction

Learning discrete representations of images enables autoregressive (AR) models to achieve promising results on image generation. Here, an image is encoded into a feature map, which is represented as a sequence of discrete codes [13, 33] or code stacks [23]. Then, an AR model generates a sequence of codes in the raster scan order and decodes the codes into an image. Consequently, AR models show high performance and scalability on large-scale datasets [13, 23, 26].

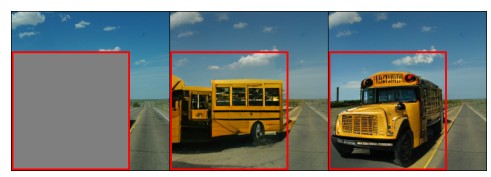

Figure 1: Examples of image inpainting by an AR model (middle) and ours (right).

Despite the promising results of AR models, we postulate that the ability of AR models is limited due to the lack of considering global contexts in the generation process. Specifically, since AR models generate images by sequentially predicting the

---

[*]Equal contribution
[†]Corresponding author

36th Conference on Neural Information Processing Systems (NeurIPS 2022).

next code and attending to only precedent codes generated, they neither exploit the later part of the generated image nor consider the global contexts during generation. For example, Figure 1 (middle) shows that an AR model fails to generate a coherent image, when it is asked to inpaint the masked region of Figure 1 (left) with a school bus. Such a failure is due to the inability of AR models to refer to the context of traffic lane on the right side of the masked region.

To address this issue, we propose an effective image generation framework, *Draft-and-Revise*, with a contextual transformer to exploit the global contexts of images. Given a randomly masked image, the contextual transformer is first trained to infill the masks by bidirectional self-attentions similarly to BERT [8]. To fully leverage the contextual prediction in generation, we propose Draft-and-Revise decoding which has two phases, *draft* and *revise*, imitating the image generation process of a human expert who draws a draft first and iteratively revises the draft to improve its quality. In the draft phase, the model first infills an empty image to generate a draft image with diverse contents despite the rather low-quality. In the revise phase, the visual quality of the draft is iteratively improved, while the global contexts of the draft are preserved and exploited. Consequently, our *Draft-and-Revise* with contextual transformer effectively generates high-quality images with diverse contents.

We use residual-quantized VAE (RQ-VAE) [23] to implement our image generation framework, since RQ-VAE generalizes vector-quantized VAE (VQ-VAE) [33] by representing an image as a sequence of code stacks instead of a sequence of codes. Then, we propose Contextual RQ-Transformer as a contextual transformer for masked code stack modeling of RQ-VAE. Specifically, given a sequence of randomly masked code stacks, Contextual RQ-Transformer first uses a bidirectional transformer to capture the global contexts of unmasked code stacks. Based on the global contexts, the masked code stacks are predicted in parallel, while the codes in each masked code stack are sequentially predicted. In experiments, our Draft-and-Revise framework with Contextual RQ-Transformer achieves state-of-the-art results on conditional image generation and remarkable improvements on image inpainting. In addition, we demonstrate that Draft-and-Revise decoding can effectively control the quality-diversity trade-off in image generation to achieve high performance.

The main contributions of this paper are summarized as follows. 1) We propose an intuitive and powerful framework, *Contextual RQ-Transformer*, for masked code stack modeling of RQ-VAE based on a bidirectional transformer. 2) We propose a novel two-phase decoding, *Draft-and-Revise*, for bidirectional transformers to fully exploit the global contexts during image generation and achieve state-of-the-art results on class- and text-conditional image generation benchmarks. 3) An extensive ablation study validates the effectiveness of Draft-and-Revise decoding on controlling the quality-diversity trade-off and its capability to generate high-quality images with diverse contents.

## 2 Related Work

**Discrete Representation for Image Generation**    By representing an image as a sequence of codes, VQ-VAE [33] becomes an important part for high-resolution image generation [6, 10, 15, 23, 26, 33], but suffers from low quality of reconstructed images. However, VQGAN [13] significantly improves the perceptual quality of reconstructed images by adding the adversarial and perceptual losses into the training objective of VQ-VAE. As a generalized approach of VQ-VAE and VQGAN, RQ-VAE [23] represents an image as a sequence of code stacks, which consists of ordered codes, and reduces the sequence length, while preserving the reconstruction quality. Then, RQ-Transformer [23] achieves high performance with lower computational costs on generating high-resolution images. However, as an AR model of RQ-VAE, RQ-Transformer cannot capture the global contexts of generated images.

**Generation Tasks with Bidirectional Transformers**    To overcome the limitation of AR models on unidirectional architecture, bidirectional transformers have been studied for generative tasks. After a bidirectional transformer is trained to infill a random mask as the masked token modeling of BERT [8], an effective decoding method has been proposed for bidirectional transformers to generate texts [14, 30], images [6, 35], or videos [16]. For image generation, M6-UFC [35] first incorporates the masked token modeling in the codes of VQGAN [13] for image generation. MaskGIT [6] proposes the confidence-based decoding to achieve high performance on ImageNet [7]. Similar to M6-UFC [35] and MaskGIT [6], we also incorporate the masked token modeling of RQ-VAE in a bidirectional transformer for image generation. However, instead of the confidence-based decoding [6, 35], we propose a novel two-phase decoding for a bidirectional transformer to achieve state-of-the-art performance and validate its effectiveness. Recently, discrete diffusion models [1, 4, 12, 15] also uses

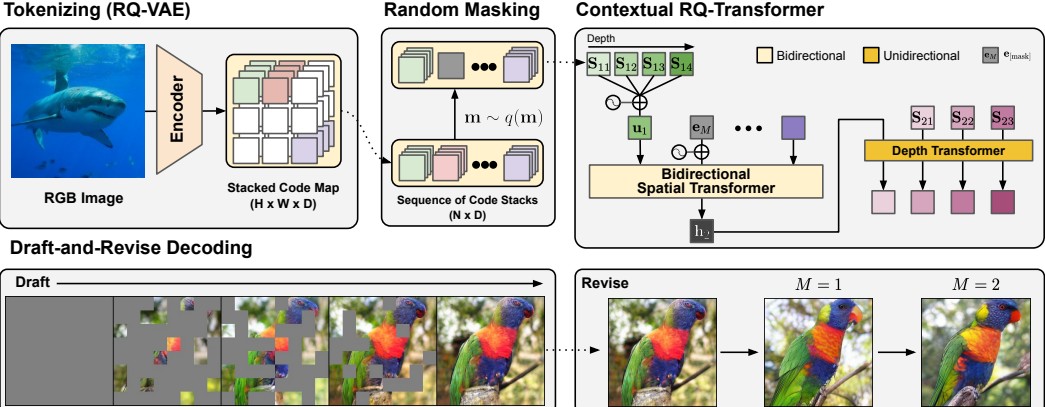

Figure 2: The overview of Draft-and-Revise framework with Contextual RQ-Transformer. Our framework exploits global contexts of images to generate high-quality images with diverse contents.

bidirectional transformers to generate an image, and the training with an absorbing state [1] is also the same to infill randomly masked sequence [4, 6]. However, different from the reverse process of diffusion models, our decoding method has explicit two phases to generate high-quality images with diverse contents.

# 3  *Draft-and-Revise* Framework for Effective Image Generation

In this section, we propose our *Draft-and-Revise* framework for effective image generation using bidirectional contexts of images. We first review RQ-VAE [23] as a generalization of VQ-VAE. Then, we propose Contextual RQ-Transformer which is trained to infill a randomly masked sequence of code stacks of RQ-VAE by understanding bidirectional contexts of unmasked parts in the sequence. Lastly, we propose *draft-and-revise* decoding for a bidirectional transformer to effectively generate high-quality images exploiting global contexts of images. Figure 2 provides the overview of our proposed framework, including Contextual RQ-Transformer and Draft-and-Revise decoding.

## 3.1  Residual-Quantized Variational Autoencoder (RQ-VAE)

RQ-VAE [23] represents an image as a sequence of code stacks. Let a codebook $\mathcal{C} = \{(k, \mathbf{e}(k))\}_{k \in [K]}$ include pairs of a code $k$ and its code embedding $\mathbf{e}(k) \in \mathbb{R}^{n_z}$, where $K = |\mathcal{C}|$ is the codebook size and $n_z$ is the dimensionality of $\mathbf{e}(k)$. Given a vector $\mathbf{z} \in \mathbb{R}^{n_z}$, $\mathcal{Q}(\mathbf{z}; \mathcal{C})$ is defined as the code of $\mathbf{z}$:

$$\mathcal{Q}(\mathbf{z}; \mathcal{C}) = \arg\min_k \|\mathbf{z} - \mathbf{e}(k)\|_2^2. \tag{1}$$

Then, RQ with depth $D$ represents a vector as a *code stack* which consists of $D$ codes:

$$\mathcal{RQ}(\mathbf{z}; \mathcal{C}, D) = (k_1, \cdots, k_D) \in [K]^D, \tag{2}$$

where $k_d$ is the $d$-th code of $\mathbf{z}$. Specifically, RQ first initializes the 0-th residual vector as $\mathbf{r}_0 = \mathbf{z}$, and then recursively discretizes a residual vector $\mathbf{r}_{d-1}$ and computes the next residual vector $\mathbf{r}_d$ as

$$k_d = \mathcal{Q}(\mathbf{r}_{d-1}; \mathcal{C}), \qquad \mathbf{r}_d = \mathbf{r}_{d-1} - \mathbf{e}(k_d), \tag{3}$$

for $d \in [D]$. Finally, $\mathbf{z}$ is approximated by the sum of the $D$ code embeddings $\hat{\mathbf{z}} := \sum_{d=1}^{D} \mathbf{e}(k_d)$. We remark that RQ is a generalized version of VQ, as RQ with $D = 1$ is equivalent to VQ. For $D > 1$, RQ conducts a finer approximation of $\mathbf{z}$ as the quantization errors are sequentially reduced as $d$ increases. Here, the coarse-to-fine approximation ensures the $D$ codes to be sequentially dependent.

RQ-VAE represents an image as a map of code stacks. Specifically, a given image $\mathbf{X}$ is first converted to a low-resolution feature map $\mathbf{Z} = E(\mathbf{X}) \in \mathbb{R}^{H \times W \times n_z}$, and then each feature vector $\mathbf{Z}_{hw}$ at spatial position $(h, w)$ is discretized into a code stack by RQ with depth $D$. As a result, we get a map of code stacks $\mathbf{S} \in [K]^{H \times W \times D}$. Further details of RQ-VAE are referred to Appendix.

## 3.2 Contextual Transformer for Image Generation with Global Contexts

As a bidirectional transformer for RQ-VAE, we propose Contextual RQ-Transformer for image generation based on a contextual understanding of images. First, we adopt the pretraining of BERT [8] to formulate a masked code stack modeling of RQ-VAE. Then, we introduce how Contextual RQ-Transformer infills the randomly masked code stacks after reading the given contextual information.

### 3.2.1 Masked Code Stack Modeling of RQ-VAE

By adopting the pretraining of BERT [8], we formulate the masked code stack modeling of RQ-VAE with a contextual transformer to generate an image by iterative mask-infilling as non-AR models [14]. We first convert the map $\mathbf{S} \in [K]^{H \times W \times D}$ into a sequence of code stacks $\mathbf{S}' \in [K]^{N \times D}$ using the raster-scan ordering, where $N = HW$ and $\mathbf{S}'_n = (\mathbf{S}'_{n1}, \cdots, \mathbf{S}'_{nD}) \in [K]^D$ for $n \in [N]$. We denote $\mathbf{S}'$ as $\mathbf{S}$ for the brevity of notation. A mask vector $\mathbf{m}$ is defined as a binary vector $\mathbf{m} \in \{0, 1\}^N$ to indicate the spatial positions to be masked. Then, the masked sequence $\mathbf{S}_{\backslash \mathbf{m}}$ of $\mathbf{S}$ by $\mathbf{m}$ is defined as

$$(\mathbf{S}_{\backslash \mathbf{m}})_n = \begin{cases} \mathbf{S}_n & \text{if } \mathbf{m}_n = 0 \\ [\text{MASK}]^D & \text{if } \mathbf{m}_n = 1 \end{cases}, \tag{4}$$

where [MASK] is a mask token to substitute for $\mathbf{S}_{nd}$ if $\mathbf{m}_n = 1$. Given a random mask vector $\mathbf{m} \sim q(\mathbf{m})$, the masked code stacks given $\mathbf{S}_{\backslash \mathbf{m}}$ are modeled as

$$\prod_{n:\mathbf{m}_n=1} p(\mathbf{S}_n | \mathbf{S}_{\backslash \mathbf{m}}) = \prod_{n:\mathbf{m}_n=1} \prod_{d=1}^{D} p(\mathbf{S}_{nd} | \mathbf{S}_{n,<d}, \mathbf{S}_{\backslash \mathbf{m}}), \tag{5}$$

where $q(\mathbf{m})$ is a mask distribution where the masking portion $\sum_{n=1}^{N} \mathbf{m}_i / N$ in $(0, 1]$ as well as the masking positions are randomly chosen. Instead of fixing the portion to 15% as in BERT, training a model with a random masking portion from $(0, 1]$ enables the model to generate new images based on various masking patterns including $\mathbf{m}_n = 1$ for all $n$. We explain the details of $q(\mathbf{m})$ in Section 3.2.3.

The left-hand side of Eq. 5 implies that all masked code stacks can be decoded in parallel based on the contexts of $\mathbf{S}_{\backslash \mathbf{m}}$. If $D = 1$, Eq. 5 becomes equivalent to conventional masked token modeling of texts [8] and images [6, 16] where a single token at each masked position is predicted. For $D > 1$, the $D$ codes of $\mathbf{S}_n$ are autoregressively predicted, as they are sequentially computed in Eq. 3 for a coarse-to-fine approximation and hence well-suited for an AR prediction. We show that the effectiveness of our framework is generalizable regardless of $D$ in Appendix B. In addition, the parallel decoding can control the trade-off between the quality and sampling speed of image generation.

### 3.2.2 Contextual RQ-Transformer

We modify the previous RQ-Transformer [23] for masked code stack modeling with bidirectional contexts in Eq. 5. Contextual RQ-Transformer consists of *Bidirectional Spatial Transformer* and *Depth Transformer*: Bidirectional Spatial Transformer understands contextual information in the unmasked code stacks using bidirectional self-attentions, and Depth Transformer infills the masked code stacks in parallel, by autoregressively predicting the $D$ codes at each position.

**Bidirectional Spatial Transformer**  Given a masked sequence of code stacks $\mathbf{S}_{\backslash \mathbf{m}}$, bidirectional spatial transformer first embeds the masked sequence $\mathbf{S}_{\backslash \mathbf{m}}$ using the code embeddings of RQ-VAE as

$$\mathbf{u}_n = \text{PE}_N(n) + \begin{cases} \sum_{d=1}^{D} \mathbf{e}(\mathbf{S}_{nd}) & \text{if } \mathbf{m}_n = 0 \\ \mathbf{e}_{[\text{MASK}]} & \text{if } \mathbf{m}_n = 1 \end{cases}, \tag{6}$$

where $\text{PE}_N(n)$ is an embedding for position $n$, and $\mathbf{e}_{[\text{MASK}]} \in \mathbb{R}^{n_z}$ is a mask embedding. Then, the bidirectional self-attention blocks, $f_\theta^{\text{spatial}}$, extracts the context vector $\mathbf{h}_n$ to predict a code stack $\mathbf{S}_n$,

$$(\mathbf{h}_1, \cdots, \mathbf{h}_N) = f_\theta^{\text{spatial}}(\mathbf{u}_1, \cdots, \mathbf{u}_N). \tag{7}$$

**Depth Transformer**  The code stack at a masked position $\mathbf{S}_n = (\mathbf{S}_{n1}, \cdots, \mathbf{S}_{nD})$ is autoregressively predicted. The input of depth transformer $(\mathbf{v}_{nd})_{d=1}^{D}$ is defined as

$$\mathbf{v}_{nd} = \text{PE}_D(d) + \begin{cases} \mathbf{h}_n & \text{if } d = 1 \\ \sum_{d'=1}^{d-1} \mathbf{e}(\mathbf{S}_{nd'}) & \text{if } d > 1 \end{cases} \tag{8}$$

---

**Algorithm 1** UPDATE of **S**

---

**Require:** A sequence of code stacks **S**, a partition $\mathbf{\Pi} = (\mathbf{m}^1, \cdots, \mathbf{m}^T)$, a model $\theta$
 1: **for** $t = 1, \cdots, T$ **do**
 2:     Sample $\mathbf{S}_n \sim p_\theta(\mathbf{S}_n|\mathbf{S}_{\backslash \mathbf{m}^t})$   $\forall n : \mathbf{m}_n^t = 1$            ▷ update the codes at masked positions
 3: **end for**
 4: **return S**

---

---

**Algorithm 2** Draft-and-Revise decoding

---

**Require:** Partition sampling distributions $p_{\text{draft}}$ and $p_{\text{rev}}$, the number of revision iterations $M$
     `/* draft phase */`
 1: $\mathbf{S}^{\text{empty}} \leftarrow ([\text{MASK}], \cdots, [\text{MASK}])^N$               ▷ initialize empty code map
 2: Sample $\mathbf{\Pi} \sim p(\mathbf{\Pi}; T_{\text{draft}})$
 3: $\mathbf{S}^{\text{draft}} \leftarrow \text{UPDATE}(\mathbf{S}^{\text{empty}}, \mathbf{\Pi}; \theta)$            ▷ generate a draft code map
     `/* revision phase */`
 4: $\mathbf{S}^0 \leftarrow \mathbf{S}^{\text{draft}}$
 5: **for** $m = 1, \cdots, M$ **do**
 6:     Sample $\mathbf{\Pi} \sim p(\mathbf{\Pi}; T_{\text{revise}})$
 7:     $\mathbf{S}^m \leftarrow \text{UPDATE}(\mathbf{S}^{m-1}, \mathbf{\Pi}; \theta)$         ▷ iteratively revise the code map
 8: **end for**
 9: **return** $\mathbf{S}^M$

---

where $\text{PE}_D(d)$ is the positional embedding for depth $d$. Then, depth transformer $f_\theta^{\text{depth}}$, which consists of causal attention blocks, outputs the logit $\mathbf{p}_{nd}$ to predict the $d$-th code $\mathbf{S}_{nd}$ at position $n$ as

$$\mathbf{p}_{nd} = f_\theta^{\text{depth}}(\mathbf{v}_{n1}, \cdots, \mathbf{v}_{nd}) \quad \text{and} \quad p_\theta(\mathbf{S}_{nd} = k|\mathbf{S}_{n,<d}, \mathbf{S}_{\backslash \mathbf{m}}) = \text{softmax}(\mathbf{p}_{nd})_k. \tag{9}$$

We remark that the architecture of Contextual RQ-Transformer subsumes bidirectional transformers. Specifically, RQ-Transformer with the depth $D = 1$ is equivalent to a bidirectional transformer since the depth transformer becomes a multilayer perceptron with layer normalization [2].

### 3.2.3 Training of Contextual RQ-Transformer

For the training of Contextual RQ-Transformer, let us define a mask distribution $q(\mathbf{m})$ with a mask scheduling function $\gamma$. Following previous approaches [6, 14, 16], the scheduling function $\gamma$ is chosen to be decreasing and to satisfy $\gamma(0) = 1$ and $\gamma(1) = 0$. Then, a mask $\mathbf{m} \sim q(\mathbf{m})$ is specified as

$$r \sim \text{Unif}([0, 1)) \quad \text{and} \quad \mathbf{m} \sim \text{Unif}(\{\mathbf{m} : |\mathbf{m}| = \lceil \gamma(r) \cdot N \rceil\}), \tag{10}$$

where $|\mathbf{m}| = \sum_{n \in [N]} \mathbf{m}_n$ is the count of masked positions. Finally, the training objective of Contextual RQ-Transformer is to minimize the negative log-likelihood of masked code stacks:

$$\mathcal{L} = \mathbb{E}_{\mathbf{m} \sim q(\mathbf{m})} \left[ \mathbb{E}_{\mathbf{S}} \left[ \sum_{n:\mathbf{m}_n=1} \sum_{d=1}^{D} -\log p_\theta(\mathbf{S}_{nd}|\mathbf{S}_{n,<d}, \mathbf{S}_{\backslash \mathbf{m}}) \right] \right]. \tag{11}$$

### 3.3 Draft-and-Revise: Two-Phase Decoding with Global Contexts of Generated Images

We propose a novel decoding algorithm, *Draft-and-Revise*, for Contextual RQ-Transformer to effectively generate high-quality images with diverse visual contents. We remark that our effective decoding method is required for bidirectional transformers to fully exploit the global contexts of images and achieve high performance of image generation. Here, we introduce the details of our Draft-and-Revise decoding and then explain how the two-phase decoding of Draft-and-Revise can effectively control the quality-diversity trade-off of generated images.

We define a partition $\mathbf{\Pi} = (\mathbf{m}^1, \cdots, \mathbf{m}^T)$ as a collection of pairwise disjoint $T$ mask vectors to cover all spatial positions, where $\sum_{t=1}^{T} \mathbf{m}_n^t = 1$ for all $n \in [N]$. A partition $\mathbf{\Pi}$ is sampled from the distribution $p(\mathbf{\Pi}; T)$, which is the uniform distribution over all balanced partitions with size $T$:

$$p(\mathbf{\Pi}; T) = \text{Unif}\left(\{\mathbf{\Pi} = (\mathbf{m}^1, \cdots, \mathbf{m}^T) : |\mathbf{m}^t| = \tfrac{N}{T} \; \forall t \in [T]\}\right). \tag{12}$$

Table 1: FIDs, ISs, Precisions, and Recalls for class-conditional generation on ImageNet [7]. † denotes the use of pretrained classifier for rejection sampling, gradient guidance, or training.

| | Params | $H \times W \times D$ | FID↓ | IS↑ | Precision↑ | Recall↑ |
|---|---|---|---|---|---|---|
| BigGAN-deep [5] | 112M | - | 6.95 | 202.6 | 0.87 | 0.23 |
| StyleGAN-XL† [29] | 166M | - | 2.3 | 262.1 | 0.78 | 0.53 |
| ADM [9] | 554M | - | 10.94 | 101.0 | 0.69 | 0.63 |
| ADM-G† [9] | 608M | - | 4.59 | 186.7 | 0.82 | 0.52 |
| ImageBART [12] | 3.5B | $16\times16\times1$ | 21.19 | 61.6 | - | - |
| VQ-Diffusion [15] | 518M | $16\times16\times1$ | 11.89 | - | - | - |
| LDM-8 [27] | 395M | $32\times32$ | 15.51 | 79.03 | 0.65 | 0.63 |
| LDM-8-G† [27] | 506M | $32\times32$ | 7.76 | 209.52 | 0.84 | 0.35 |
| MaskGIT [6] | 227M | $16\times16\times1$ | 6.18 | 182.1 | 0.80 | 0.51 |
| VQ-GAN [13] | 1.4B | $16\times16\times1$ | 15.78 | 74.3 | - | - |
| RQ-Transformer [23] | 1.4B | $8\times8\times4$ | 8.71 | 119.0 | 0.71 | 0.58 |
| RQ-Transformer [23] | 3.8B | $8\times8\times4$ | 7.55 | 134.0 | 0.73 | 0.58 |
| RQ-Transformer† [23] | 3.8B | $8\times8\times4$ | 3.80 | 323.7 | 0.82 | 0.50 |
| **Contextual RQ-Transformer** | 333M | $8\times8\times4$ | 5.45 | 172.6 | 0.81 | 0.49 |
| **Contextual RQ-Transformer** | 821M | $8\times8\times4$ | 3.45 | 221.9 | 0.82 | 0.52 |
| **Contextual RQ-Transformer** | 1.4B | $8\times8\times4$ | 3.41 | 224.6 | 0.79 | 0.54 |
| Validation Data | - | - | 1.62 | 234.0 | 0.75 | 0.67 |

We first define a procedure UPDATE($\mathbf{S}, \mathbf{\Pi}$) to update the sequence $\mathbf{S}$ as described in Algorithm 1, which updates $\mathbf{S}_n$ with $\mathbf{m}_n^t = 1$ for $t \in [T]$. Then, *Draft-and-Revise* decoding in Algorithm 2 generates a draft from the empty sequence of code stacks and improves the quality of the draft.

**Draft phase** In the draft phase, our model gradually infills the empty sequence of code stacks to generate a draft image, considering the global contexts of infilled code stacks. Let $\mathbf{S}^{\text{empty}}$ be an empty sequence of code stacks with $\mathbf{S}_n^{\text{empty}} = [\text{MASK}]^D$ for all $n$. Given a partition size $T_{\text{draft}}$, our model generates a draft image as

$$\mathbf{S}^{\text{draft}} = \text{UPDATE}(\mathbf{S}^{\text{empty}}, \mathbf{\Pi}; \theta) \quad \text{where} \quad \mathbf{\Pi} \sim p(\mathbf{\Pi}; T_{\text{draft}}). \quad (13)$$

**Revise phase** The generated draft $\mathbf{S}^{\text{draft}}$ is repeatedly revised to improve the visual quality of the image, while preserving the overall structure of the draft. Given a partition size $T_{\text{revise}}$ and the number of updates $M$, the draft $\mathbf{S}^0 = \mathbf{S}^{\text{draft}}$ is repeatedly updated $M$ times as

$$\mathbf{S}^m = \text{UPDATE}(\mathbf{S}^{m-1}, \mathbf{\Pi}; \theta) \quad \text{where} \quad \mathbf{\Pi} \sim p(\mathbf{\Pi}; T_{\text{revise}}) \quad \text{for } m = 1, \cdots, M. \quad (14)$$

Note that Draft-and-Revise is not a tailored method, since we can adopt any mask-infilling-based generation method [4, 6] for UPDATE in Algorithm 1. For example, confidence-based decoding [6, 16], which iteratively updates $\mathbf{S}$ from high-confidence to low-confidence predictions, can be used for UPDATE. However, we find that confidence-based decoding generates low-diversity images with oversimplified contents, since a model tends to predict simple visual patterns with high confidence. In addition, confidence-based decoding often leads to biased unmasking patterns, which are not used in training. We attach the detailed discussion about confidence-based decoding in Appendix C. Thus, we use a uniformly random partition $\mathbf{\Pi}$ in UPDATE as the most simplified rule, leaving investigations on sophisticated update methods as future work.

We postulate that our Draft-and-Revise can generate high-quality images with diverse contents by explicitly dividing two phases. Specifically, a model first generates draft images with diverse visual contents despite the rather low quality of drafts. After semantically diverse images are generated as drafts, we use sampling strategies such as temperature scaling [19] and classifier-free guidance [20] in the revise phase to improve the visual quality of the drafts, while preserving the major semantic contents in drafts. Thus, our method can improve the performance of image generation by effectively controlling the quality-diversity trade-off. In addition, we emphasize that the two-phased decoding is intuitive and resembles the image generation process of human experts, who repeatedly refine their works to improve the quality after determining the overall contents first.

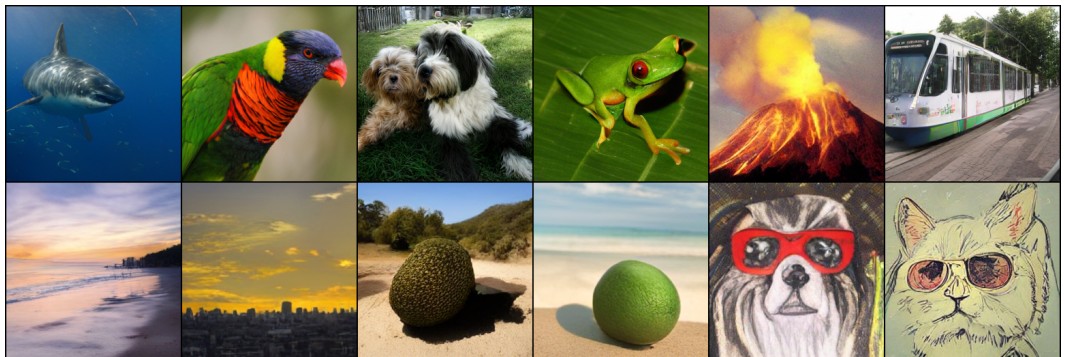

Figure 3: The examples of generated 256×256 images of our model trained on (Top) ImageNet and (Bottom) CC-3M. The used text conditions are "Sunset over the skyline of a {beach, city}.", "an avocado {in the desert, on the seashore}.", and "a painting of a {dog, cat} with sunglasses.".

## 4  Experiments

In this section, we show that our Draft-and-Revise with Contextual RQ-Transformer can outperform previous approaches for class- and text-conditional image generation. In addition, we conduct an extensive ablation study to understand the effects of Draft-and-Revise decoding on the quality and diversity of generated images, and the sampling speed. We use the publicly released RQ-VAE [23] to represent a 256×256 resolution of images as $8\times8\times4$ codes. For a fair comparison, we make Contextual RQ-Transformer have the same model size as the previous RQ-Transformer [23]. For training, the quarter-period of cosine is used as the mask scheduling function $\gamma$ in Eq. 10 following the previous studies [6, 24]. We include the implementation details in Appendix.

### 4.1  Class-conditional Image Generation

We train Contextual RQ-Transformer with 333M, 821M, and 1.4B parameters on ImageNet [7] for class-conditional image generation. For Draft-and-Revise decoding, we use $T_{\text{draft}} = 64$, $T_{\text{revise}} = 2$, and $M = 2$. We use temperature scaling [19] and classifier-free guidance [20] only in the revise phase, while none of the strategies are applied in the draft phase. Fréchet Inception Distance (FID) [18], Inception Score (IS) [28], and Precision and Recall [22] are used for evaluation measures.

Table 1 shows that Contextual RQ-Transformer significantly outperforms the previous approaches. Notably, Contextual RQ-Transformer with 333M parameters outperforms RQ-Transformers with 1.4B and 3.8B parameters on all evaluation measures, despite having only about 4.2× and 11.4× fewer parameters. In addition, the performance is improved as the number of parameters increases to 821M and 1.4B. Our model with 333M parameters is competitive with MaskGIT [6], and hence we conduct additional analysis on the effect of confidence-based decoding in Appendix C. Contextual RQ-Transformer can achieve the lower FID score without a pretrained classifier than ADM-G and 3.8B parameters of RQ-Transformer with the use of pretrained classifier. StyleGAN-XL also uses a pretrained classifier during both training and image generation and achieves the lowest FID in Table 1. However, our model with 1.4B parameters has higher precision and recall than StyleGAN-XL, implying that our model generates images of better fidelity and diversity without a pretrained classifier. Our high performance without a classifier is remarkable, since the gradient guidance and rejection sampling are the tailored techniques to the model-based evaluation metrics in Table 1. Considering that the performance is marginally improved as the number of parameters increases from 821M to 1.4B, an improved RQ-VAE can boost the performance of Contextual RQ-Transformer, since the reconstruction quality determines the best results of generated images.

### 4.2  Text-conditional Image Generation

We train our model with 333M and 654M parameters on CC-3M [32] for text-to-image (T2I) generation as described in Appendix A. A text condition is encoded into 32 tokens using Byte Pair Encoding [31, 34]. We report CLIP-score [25] with ViT-B/32 [11] as the correspondence between texts and images.

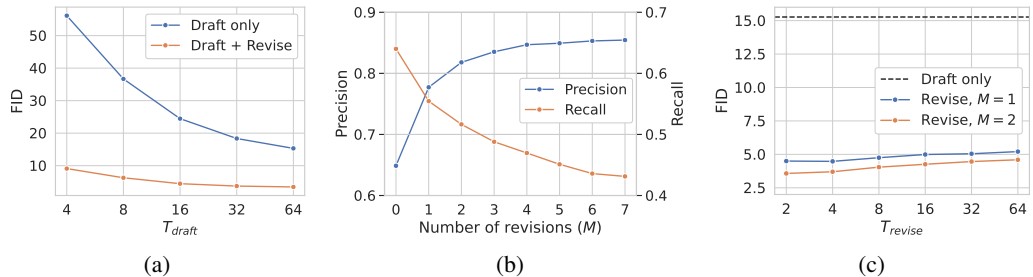

Figure 4: Ablation study of Draft-and-Revise decoding. (a) FIDs with the partition sizes of the draft phase $T_{\mathrm{draft}}$. (b) Precision and recall with the revision iterations $M$. (c) FID with $T_{\mathrm{revise}}$.

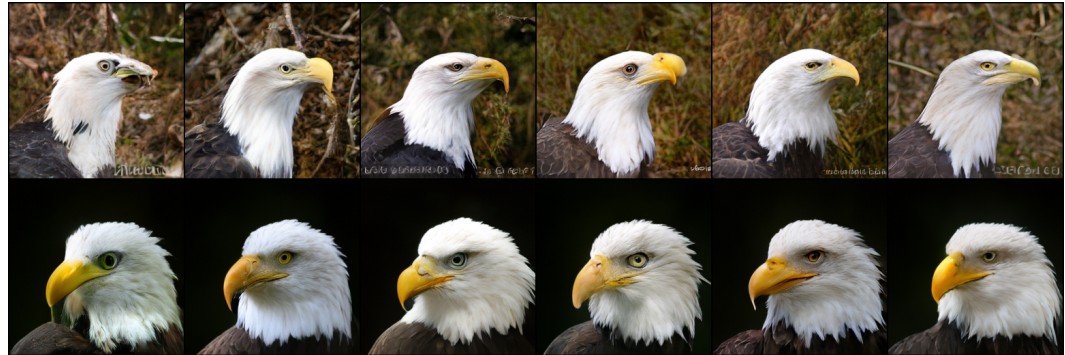

Figure 5: Examples of generated images in the draft phase (left) and revise phases at each iteration $M = 1, 2, 3, 4, 5$. The draft images are generated with $T_{\mathrm{draft}} = 8$ (top) and $T_{\mathrm{draft}} = 64$ (bottom).

Contextual RQ-Transformer in Table 2 outperforms the previous T2I generation models. Contextual RQ-Transformer with 333M parameters achieves better FID than RQ-Transformer with 654M parameters, and outperforms ImageBART and LDM-4, although our model has $12\times$ fewer parameters than ImageBART. When we increase the number of parameters to 654M, our model achieves state-of-the-art FID on CC-3M. In Figure 3, our model generates images with unseen texts in CC-3M.

Table 2: FIDs and CLIP scores [25] on the validation dataset of CC-3M [32] for T2I generation.

|  | Params | FID↓ | CLIP-s↑ |
|---|---|---|---|
| VQ-GAN [13] | 600M | 28.86 | 0.20 |
| ImageBART [12] | 2.8B | 22.61 | 0.23 |
| LDM-4 [27] | 645M | 17.01 | 0.24 |
| RQ-Transformer [23] | 654M | 12.33 | 0.26 |
| **Ours** | 333M | 10.44 | 0.26 |
| **Ours** | 654M | 9.80 | 0.26 |

### 4.3 Conditional Image Inpainting

We conduct conditional image inpainting where a model infills a masked area according to the given condition and contexts. Figure 1 shows the example of image inpainting by RQ-Transformer (middle) and Contextual RQ-Transformer (right), when the class-condition is *school bus*. RQ-Transformer cannot attend to the right and bottom sides of the masked area and fails to generate a coherent image with given contexts. However, our model can complete the image to be coherent with given contexts by exploiting global contexts. We attach more examples of image inpainting in Appendix.

### 4.4 Ablation Study on Draft-and-Revise

We conduct an extensive ablation study to demonstrate the effectiveness of Draft-and-Revise decoding of our framework. We use Contextual RQ-Transformer with 821M parameters trained on ImageNet.

**Quality improvement of draft images in the revise phase** Figure 4(a) shows the effects of $T_{\mathrm{draft}}$ on draft images and their quality improvement in the revised phase with $T_{\mathrm{revise}} = 2$ and $M = 2$.

In the draft phase, FID is improved as $T_\text{draft}$ increases from 4 to 64. At each inference, Contextual RQ-Transformer generates $N/T_\text{draft}$ code stacks in parallel, starting with the empty sequence. Thus, the model with a large $T_\text{draft}$ generates a small number of code stacks at each inference and can avoid generating incoherent code stacks in the early stage of the draft phase. Although FIDs in the draft phase are worse due to low precision, we intend to increase the diversity of generated images, since the precision is significantly improved in the revise phase as shown in Figure 4(b).

**Effect of revision iteration $M$ and partition size $T_\text{revise}$ in the revise phase**   Figure 4(b) shows the effects of the number of updates $M$ in the revise phase on the quality and diversity of generated images. Since the quality-diversity trade-off exists as the updates are repeated, we select $M = 2$ as the default hyperparameter to balance the precision and recall, considering that the increase of precision starts to slow down. Interestingly, Figure 5 shows that the overall contents remain unchanged even after $M > 2$. Thus, we claim that Draft-and-Revise decoding does not harm the perceptual diversity of generated images throughout the revise phase despite the consistent deterioration of recall.

Figure 4(c) shows the effects of $T_\text{revise}$ on the quality of generated images. The FIDs are significantly improved in the revise phase regardless of the choice of $T_\text{revise}$, but increasing $T_\text{revise}$ slightly deteriorates FIDs. We remark that some code stacks of a draft can be erroneous due to its low quality, and a model with large $T_\text{revise}$ slowly updates a small number of code stacks at once in the revise phase. Therefore, the updates with large $T_\text{revise}$ can be more influenced by the erroneous code stacks. Although $T_\text{revise} = 2$ updates half of an image at once, our draft-and-revise decoding successfully improves the quality of generated images, while preserving the global contexts of drafts, as shown in Figure 5. The study on self-supervised learning [17] also reports similar results, where a masked auto-encoder reconstructs the global contexts of an image after masking half of the image.

**Quality-diversity control of Draft-and-Revise**   Our *Draft-and-Revise* decoding can effectively control the quality-diversity trade-off in generated images. Table 3 shows FID, precision (P), and recall (R) according to the use of classifier-free guidance [20] with a scale of 1.8, while applying temperature scaling with 0.8 only to the revise phase. Contextual RQ-Transformer without the guidance already outperforms RQ-Transformer with 3.8B parameters and demonstrates the effectiveness of our framework. When the guidance is used for both draft and revise phases, the precision dramatically increases but the recall decreases to 0.33. Consequently, FID becomes worse due to the lack of diversity in generated images. However, when the guidance is applied only to the revise phase, our model achieves the lowest FID, as the quality and diversity are well-balanced. Thus, the explicitly separated two phases of Draft-and-Revise can effectively control the issue of quality-diversity trade-off by generating diverse drafts and then improving their quality.

Table 3: The effects of classifier-free guidance on the image generation.

| Draft | Revise | FID | P | R |
|---|---|---|---|---|
| | | 5.78 | 0.72 | 0.58 |
| | ✓ | 3.45 | 0.82 | 0.52 |
| ✓ | ✓ | 8.90 | 0.92 | 0.33 |

**Trade-off between quality and sampling speed**   Table 4 shows that our framework can control the trade-off between the quality and sampling speed according to $T_\text{draft}$ after we fix $T_\text{revise} = 2$ and $M = 2$. We generate 5,000 samples with batch size of 100 as the previous study [23]. Contextual RQ-Transformer with $T_\text{draft} = 8$ outperforms both FID and sampling speed of VQGAN and RQ-Transformer with 1.4B parameters. Although the sampling speed becomes slow with increased $T_\text{draft}$, the FID scores are consistently improved. We remark that the sampling speed with $T_\text{draft} = 64$ is about $3\times$ slower than RQ-Transformer, but our model outperforms 3.8B parameters of RQ-Transformer with rejection sampling in Table 1. The results show that our framework has inexpensive costs to generate high-quality images, considering that rejection sampling requires generating up to $20\times$ more samples.

Table 4: Comparison of FID and the sampling speed of image generation.

| | FID | s/sample |
|---|---|---|
| VQGAN | 15.78 | 0.16 |
| RQ-Transformer | 8.71 | 0.04 |
| *Contextual RQ-Transformer* | | |
| $T_\text{draft} = 8$ | 5.41 | 0.03 |
| $T_\text{draft} = 32$ | 3.73 | 0.06 |
| $T_\text{draft} = 64$ | 3.45 | 0.10 |

## 5 Conclusion

In this study, we have proposed *Draft-and-Revise* for an effective image generation framework with Contextual RQ-Transformer. After an image is represented as a sequence of code stacks, Contextual RQ-Transformer is trained to infill a randomly masked sequence. Then, Draft-and-Revise decoding is used to generate high-quality images by first generating a draft image with diverse contents and then improving its visual quality based on the global contexts of the draft. Consequently, we can achieve state-of-the-art results on ImageNet and CC-3M, demonstrating the effectiveness of our framework.

Our study has two main limitations to be further explored. Firstly, Draft-and-Revise decoding always updates all code stacks in the revise phase, although some code stacks might not need an update. In future work, a selective method can be developed to improve the efficiency of the revise phase by a sophisticated approach. Secondly, our generative model is not validated on various downstream tasks. Since masked token modeling is successful self-supervised learning for texts [8] and images [3, 17], a unified model for both generative and discriminative tasks [21] is worth exploration for future work.

## 6 Acknowledgements

This work was supported by Institute of Information & communications Technology Planning & Evaluation(IITP) grant funded by the Korea government(MSIT) (No.2018-0-01398: Development of a Conversational, Self-tuning DBMS, 35%; No.2021-0-00537: Visual Common Sense, 30%) and the National Research Foundation of Korea(NRF) grant funded by the Korea government(MSIT) (No. NRF-2021R1A2B5B03001551, 35%)

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
