# Draft-and-Revise: Effective Image Generation with Contextual RQ-Transformer (Supplementary Material)

## A Implementation Details

### A.1 Details of RQ-VAE

RQ-VAE [7] is a generalized version of VQ-VAE [8] and VQGAN [2], since RQ-VAE with $D = 1$ is equivalent to VQ-VAE or VQGAN. When $D > 1$, RQ-VAE recursively discretizes the feature map of an image for a precise approximation of the feature map using the codebook. When the codebook size of RQ is $K$, RQ with depth $D$ is as capable as VQ with $K^D$ size of a codebook, since RQ can represent at most $K^D$ clusters in a vector space. That is, if the codebook sizes are the same, RQ with $D > 1$ can approximate a feature vector more accurately than VQ. Thus, RQ-VAE can further reduce the spatial resolution of code map than VQ-VAE and VQ-GAN, and therefore outperforms previous autoregressive models with discrete representations.

Following the previous studies [2, 7, 8], the training of RQ-VAE uses the reconstruction loss, the commitment loss, the adversarial training [5], and the LPIPS perceptual loss [6]. The codebook $\mathcal{C}$ of RQ-VAE is updated using the exponential moving average during training [7, 8].

In experiments, we use the pretrained RQ-VAE, which is publicly available[1]. The RQ-VAE uses the codebook size of 16,384 to represent a 256×256 resolution of an image as 8×8×4 shape of a code map. The architecture of RQ-VAE is the same as VQGAN [2] except for adding residual blocks in the encoder and the decoder to reduce the spatial resolution of the code map more than VQGAN.

### A.2 Architecture of Contextual RQ-Transformer

Table 1 summarizes the architecture details of Contextual RQ-Transformers to be trained on ImageNet and CC-3M. Contextual RQ-Transformer consists of two compartments: *bidirectional spatial transformer* with $N_{\text{spatial}}$ self-attention blocks and *depth transformer* with $N_{\text{depth}}$ causal self-attention blocks. The dimensionality of embeddings in multi-headed self-attentions is denoted $d_{\text{model}}$, while the dimensionality for each attention head is 64.

Table 1: Architecture details of Contextual RQ-Transformer for ImageNet and CC-3M.

| Dataset | # params. | $N_{\text{spatial}}$ | $N_{\text{depth}}$ | $d_{\text{model}}$ |
|---------|-----------|------------|-----------|-----------|
| ImageNet | 371M | 24 | 4 | 1024 |
| | 821M | 24 | 4 | 1536 |
| | 1.4B | 42 | 6 | 1536 |
| CC-3M | 366M | 21 | 4 | 1024 |
| | 654M | 26 | 4 | 1280 |

### A.3 Training details

All Contextual RQ-Transformers are trained with AdamW optimizer with $\beta_1 = 0.9$, $\beta_2 = 0.95$, and weight decay 0.0001. Each model is trained for 300 epochs with the cosine learning rate schedule with the initial value of 0.0001 and the final value of 0, for both ImageNet and CC-3M. We use eight NVIDIA A100 GPUs to train the models with 821M and 1.4B parameters on ImageNet and the model with 650M parameters on CC-3M, while four GPUs are used for the models with 366M parameters. For our model with 821M and 1.4B parameters on ImageNet, the training takes at most 10 days.

_________________

[1]`https://github.com/kakaobrain/rq-vae-transformer`

36th Conference on Neural Information Processing Systems (NeurIPS 2022).

Table 2: Performance of Contextual RQ-Transformers on 16×16×1 RQ-VAE.

| | # params | $H \times W \times D$ | FID | P | R | s/sample |
|---|---|---|---|---|---|---|
| VQ-GAN [2] | 1.4B | 16×16×1 | 15.78 | - | - | - |
| RQ-Transformer [7] | 1.4B | 8×8×4 | 8.71 | 0.71 | 0.58 | 0.04 |
| Contextual VQ-Transformer | 350M | 16×16×1 | 6.44 | 0.79 | 0.47 | 0.83 |
| Contextual RQ-Transformer | 371M | 8×8×4 | 5.45 | 0.81 | 0.49 | 0.08 |

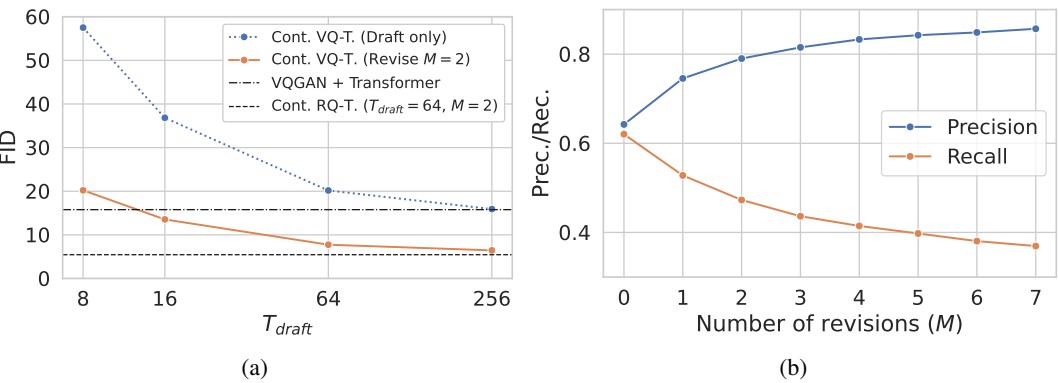

(a)                                    (b)

Figure 1: Effect of Draft-and-Revise decoding on Contextual VQ-Transformer. (a) FID subject to $T_{\text{draft}}$. (b) Precision and recall subject to $M$.

### A.4   Draft-and-Revise decoding details

We use temperature scaling with the 0.8 scale in the revise phase and do not apply the temperature scaling in the draft phase. The classifier-free guidance is also applied only to the sampling in the revise phase. We use 1.4, 1.8, and 2.0 scales of guidance for 371M, 821M, and 1.4B parameters of Contextual RQ-Transformer on ImageNet, respectively. In addition, $(M, T_{\text{revise}})$ is (3,2), (2,2), and (2,2), respectively. In the revise phase for Contextual RQ-Transformer on CC-3M, we use the 1.1 scale of classifier-free guidance with $(M, T_{\text{revise}}) = (2, 4)$.

## B   The Compatibility of Draft-and-Revise with 16×16 VQGAN

Since RQ is a generalized VQ, our framework of Draft-and-Revise with Contextual RQ-Transformer is also applicable to VQ-VAE [8] and VQGAN [2]. Note that RQ-VAE with $D = 1$ is equivalent to VQGAN, where their spatial resolutions of code maps are the same. To validate that our framework is also effective on the 16×16 shape of code map by VQGAN, we first train an RQ-VAE, which represents an image as 16×16×1 shape of code maps and has the identical architecture to VQGAN with 16×16 shape of code map. Then, we train a Contextual RQ-Transformer with 350M parameters on 16×16×1 codes. We notate the Contextual RQ-Transformer with $D = 1$ as *Contextual VQ-Transformer* throughout this section.

Figure 1 shows that the Draft-and-Revise decoding also generalizes to Contextual VQ-Transformer and can control the quality-diversity trade-off in the same manner as with the Contextual RQ-Transformer. We fix $T_{\text{revise}} = 2$ and $M = 2$, and use temperature scaling of 0.8 and classifier-free guidance of 2.4 only in the revise phase. As shown in Figure 1(a), the best performance is achieved when $T_{\text{draft}} = 256$ and the corresponding FID, precision (P), and recall (R) are 6.44, 0.79, and 0.47, respectively, as reported in Table 2. Note that Contextual VQ-Transformer outperforms the AR model such as VQGAN, although our model has 4× fewer parameters than the AR model. In addition, our draft-and-revise decoding with Contextual VQ-Transformer also works as well as Contextual RQ-Transformer, showing that the iterative updates in the revise phase consistently increase precisions and decreases recalls in Figure 1(b). Consequently, the results validate that our framework is more effective for high-quality image generation than AR modeling.

Table 3: Comparison of confidence-based mask-infilling strategies and our random partitioning strategy in the draft phase.

| | Draft Images | | | Revised Images | | |
|---|---|---|---|---|---|---|
| | FID | P | R | FID | P | R |
| $T_{\text{draft}} = 8$ | | | | | | |
| Top-C | 18.28 | 0.67 | 0.43 | 11.92 | 0.79 | 0.35 |
| Top-C-50% | 13.26 | 0.71 | 0.53 | 6.13 | 0.86 | 0.41 |
| Random | 36.68 | 0.58 | 0.60 | 6.28 | 0.80 | 0.49 |
| $T_{\text{draft}} = 64$ | | | | | | |
| Top-C | 34.07 | 0.54 | 0.42 | 27.88 | 0.62 | 0.37 |
| Top-C-50% | 7.22 | 0.75 | 0.53 | 6.84 | 0.84 | 0.42 |
| Random | 15.32 | 0.65 | 0.64 | 3.45 | 0.82 | 0.52 |

Although our framework is compatible with VQGAN, we emphasize that Contextual RQ-Transformer is more effective than Contextual VQ-Transformer. Contextual RQ-Transformer with a similar number of parameters outperforms the Contextual VQ-Transformer in terms of FID, precision, and recall in Table 2. In addition, Contextual RQ-Transformer has about $10\times$ faster speed for image generation than Contextual VQ-Transformer, since the computational complexity of self-attention is mainly determined by the sequence length. Although the comparison of FID, precision, and recall is not entirely fair due to the different performance between RQ-VAE and VQGAN, Contextual RQ-Transformer can achieve state-of-the-art performance with lower computational costs than Contextual VQ-Transformer. Thus, masked modeling in RQ representations is more effective and efficient than in VQ representations, if equipped with our Contextual RQ-Transformer.

## C  Comparison with Confidence-based Mask-infilling Strategies

In this section, we examine how the selection of UPDATE in Algorithm 1 affects the performance of image generation. First, instead of using our random updates of spatial positions, we consider a confidence-based mask-infilling strategy of MaskGIT [1] and denote the sampling strategy of MaskGIT as *Top-C*. At each inference step, *Top-C* considers the predicted confidences and determines the unmasked positions to have highly confident predictions. Then, we also consider a mixed strategy, *Top-C-50%*, which first filters out the bottom 50% confident positions, and then randomly selects the unmasked positions among the positions with top 50% high confidence. *Top-C-50%* is similar to the combination of random sampling after top-k or top-p [4] filtering. We denote our mask-infilling strategy as *Random*, which randomly determines the unmasked regions at each inference in the draft phase. We use Contextual RQ-Transformer with 821M parameters and fix the parameters of Draft-and-Revise decoding to $T_{\text{draft}} = 64$, $T_{\text{revise}} = 2$, and $M = 2$ and apply temperature scaling of 0.8 and classifier-free guidance of 1.8 in the revise phase. We report FID, precision (P), and recall (R) of the generated images in the draft and revise phases.

Table 3 shows the effect of Top-C, Top-C-50%, and Random in Draft-and-Revise decoding. Regardless of the selection of mask-infilling strategies in the draft phase, our draft-and-revise decoding consistently improves the performance of image generation after the revise phase. The results imply that our framework of Draft-and-Revise can be effectively generalized to various approaches of mask-infilling-based image generation.

When $T_{\text{draft}} = 8$, the draft images of Top-C have better FID but worse recall than the draft images of Random. Nonetheless, the quality of the draft images is significantly improved with the revise phase and subsequently Random outperforms Top-C in all three metrics. Top-C-50% achieves the best FID in both the draft and revised images, but the recall of Top-C-50% is significantly lower than the recall of Random. When $T_{\text{draft}} = 64$, the draft images of Top-C exhibit worse metrics than the draft images of Random. Although Top-C-50% achieves lower FID than Top-C and Random in terms of draft images, Random outperforms Top-C and Top-C-50% after the revise phase. This supports our claim that Draft-and-Revise decoding better controls quality-diversity trade-off when drafts are generated with maximal diversity.

By the visual analysis in Figure 2, we find that the limited performance of confidence-based mask-infilling strategies results from the bias of a model on high-confident predictions. That is, Top-C tends to predict only simple patterns in the early phase of mask-infilling, since the simple visual patterns are

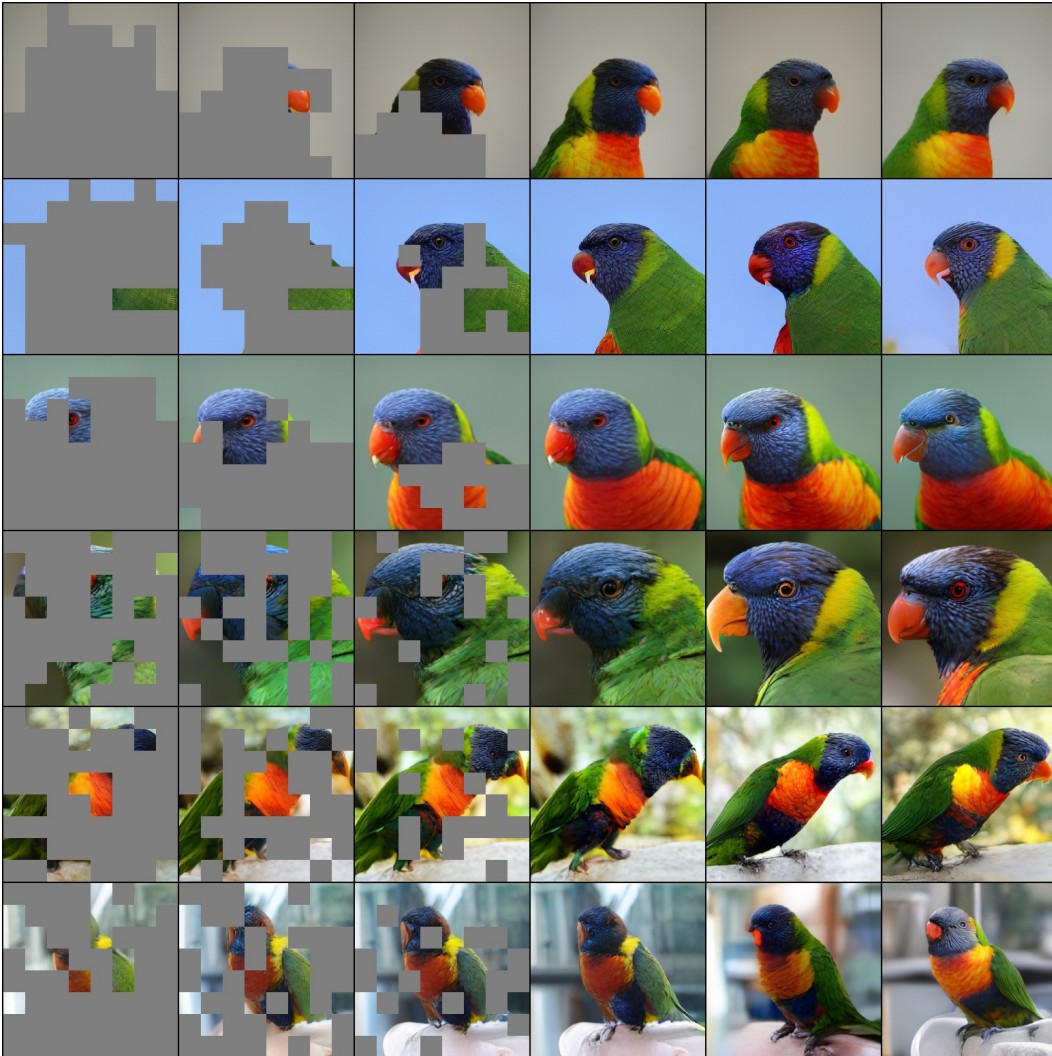

Figure 2: Intermediary samples of mask-infilling strategies with $T_{\text{draft}} = 64$ and subsequently revised samples. From left to right, the first four images are the samples with the corresponding masking pattern of every 16th mask-infilling step, and the last two images are the revised samples for $M = 1, 2$. (Top 3 rows) Mask-infilled with a confidence-based strategy, *Top-C*. (Bottom 3 rows) Mask-infilled with our strategy, *Random*.

prone to have high-confident predictions. This effect becomes more apparent when $T_{\text{draft}} = 64$ as only the code stack with the highest confidence is included in the sample at each inference, thereby the diversity of the generated images becomes severely limited. Indeed, it is shown in Figure 2 that the confidence-based methods first infill the backgrounds with simple visual patterns, and the resulting samples exhibit low diversity of visual contents. Due to the limited diversity, the confidence-based mask-infilling strategies limit the performance of FID, although our draft-and-revise decoding further improves the visual quality of generated images after the revise phase. Thus, we conclude that our sampling strategy, which first generates diverse visual contents and then improves their quality, is effective for draft-and-revise decoding to achieve high performance of image generation.

## D   Additional Generation Examples

In this section, we show additional examples of generated images by our Contextual RQ-Transformer. We use 1.4B parameters of Contextual RQ-Transformer trained on ImageNet for class-conditional

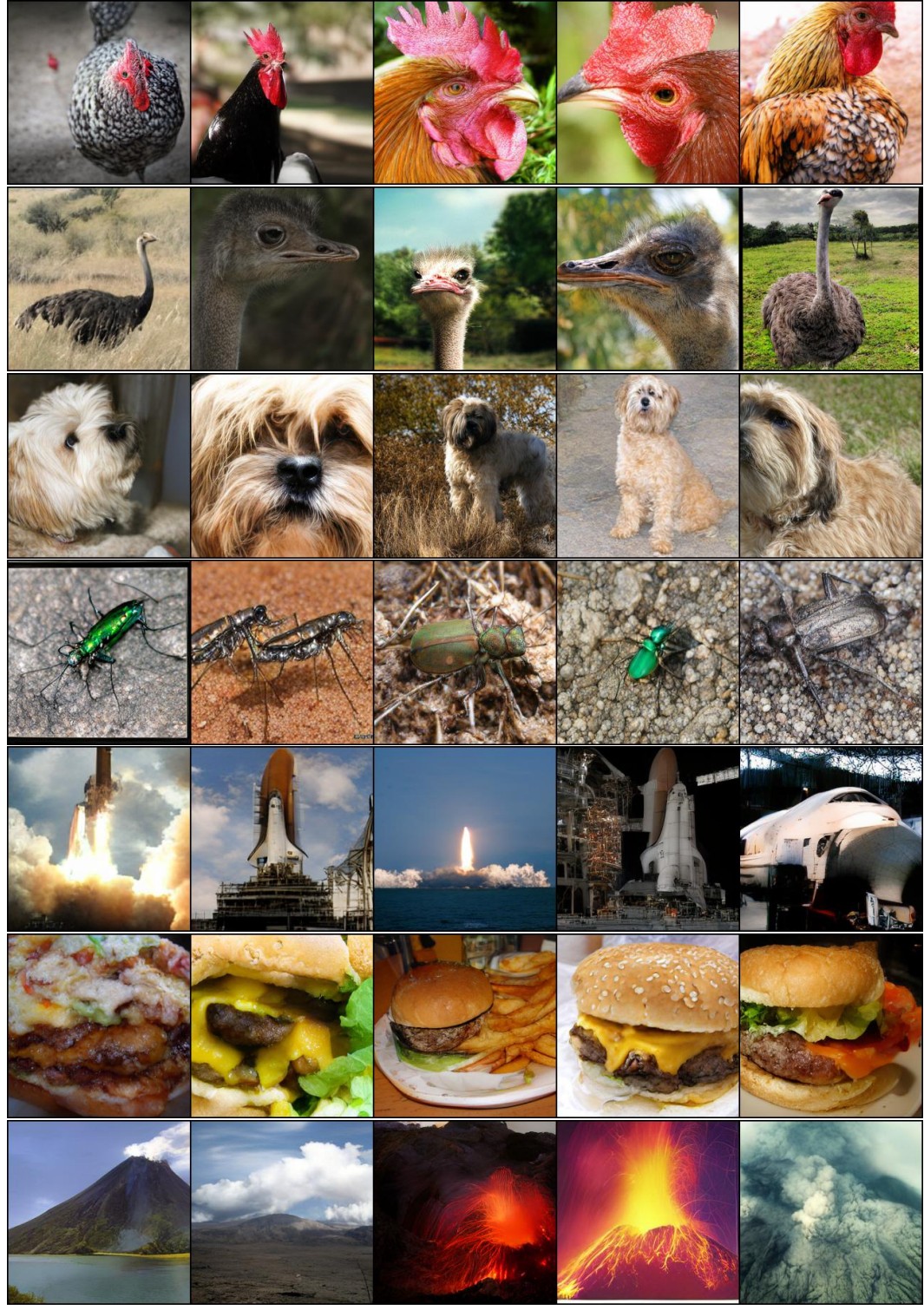

Figure 3: Additional examples of class-conditional image generation by our model with 1.4B parameters trained on ImageNet. The class conditions are Cock (7), Ostrich (9), Tibetan terrier (200), Space shuttle (812), Cheeseburger (933), and Volcano (980), respectively.

| | |
|---|---|
| *pine tree on a background of the sea*[†] | 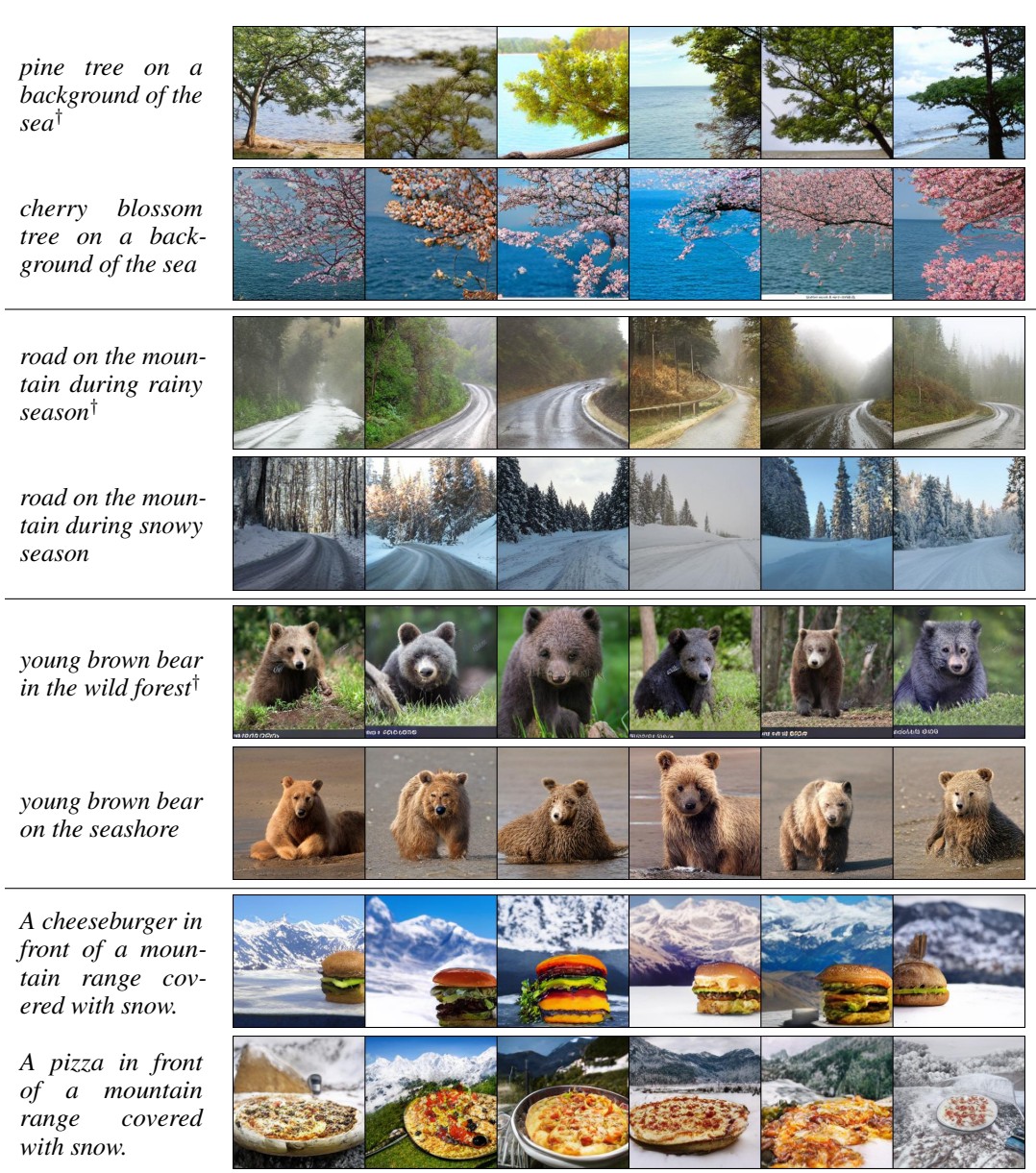 |
| *cherry blossom tree on a background of the sea* | |
| *road on the mountain during rainy season*[†] | |
| *road on the mountain during snowy season* | |
| *young brown bear in the wild forest*[†] | |
| *young brown bear on the seashore* | |
| *A cheeseburger in front of a mountain range covered with snow.* | |
| *A pizza in front of a mountain range covered with snow.* | |

Figure 4: Additional examples of text-conditional image generation by our model with 654M parameters trained on CC-3M. † denotes that the caption exists in the validation dataset of CC-3M.

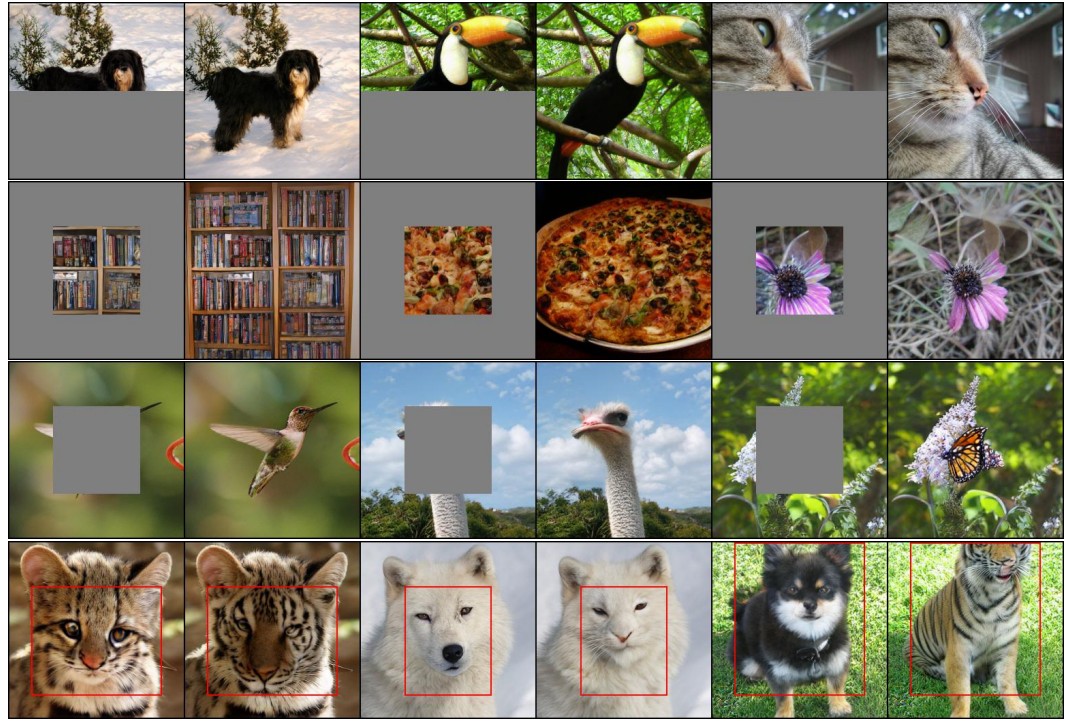

Figure 5: Additional examples of image inpainting by our model with 1.4B parameters trained on ImageNet. All masked images are taken from the validation set of ImageNet. (Top 3 rows) Inpainted images when conditioned on the class of the original image. (Bottom row) Images where the region inside the red box is inpainted with the class condition Tiger (292).

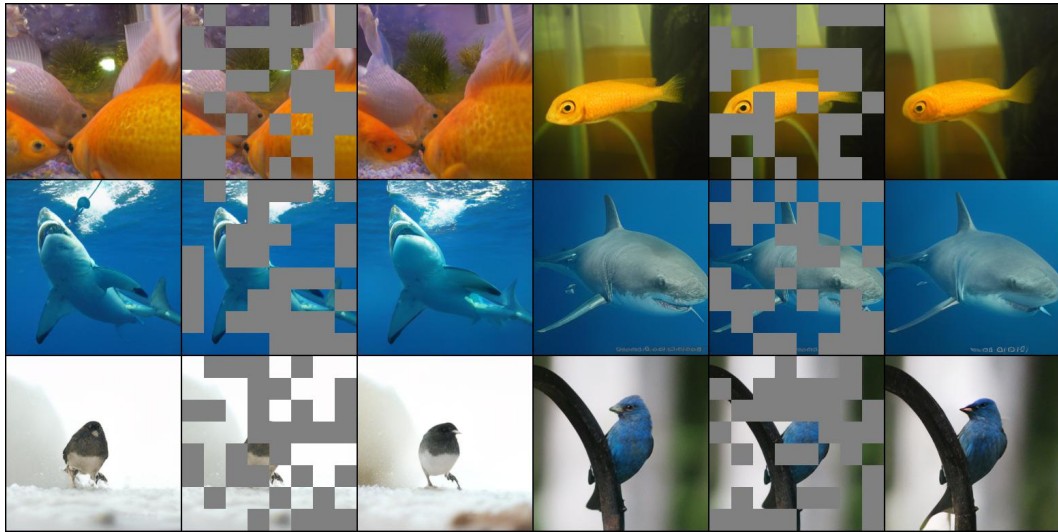

Figure 6: The triplets of the original image, randomly masked image, and mask-infilled image by our model. Although half of an image is randomly masked, Contextual RQ-Transformer can infill the masks and remains the global contexts of the original image.

image generation and 654M parameters of our model trained on CC-3M for text-conditional image generation.

### D.1 Examples of Class-conditional Image Generation

Figure 3 shows the generated images on class conditions in ImageNet. Contextual RQ-Transformer generates diverse and high-quality images conditioned on given class conditions.

### D.2 Examples of Text-conditioned Image Generation

Figure 3 shows the generated images by our model on various text conditions. We use the captions in the validation dataset[†] of CC-3M and change a keyword in the caption to validate the generalization power of our model on unseen compositions of texts. Figure 3 shows that our Contextual RQ-Transformer can generate high-quality images on diverse compositions of text conditions, even though the text condition is unseen during training.

### D.3 Examples of Image In-painting

Figure 5 shows that Draft-and-Revise decoding can also be used to inpaint the prescribed region of a given image. For image inpainting, a random partition $\Pi$ used in each application of UPDATE is set to be the partition of the masked region, instead of all spatial positions. The first three rows of Figure 5 show the image inpainting results, where the original images are taken from the validation set of ImageNet, either bottom, center, or outside is masked, and the class of the original image is given as a condition. On the other hand, the last row of Figure 5 shows that our Contextual RQ-Transformer is also capable of image editing via inpainting, by conditioning on a class-condition that is not the class of the original image.

### D.4 Examples of Mask-Infilling of Half Masked Images

After we randomly mask the half of images in the ImageNet validation dataset, Contextual RQ-Transformer infills the masked regions in Figure 6. The results show that Contextual RQ-Transformer can infill the masked regions, while preserving the global contexts of original images. Note that the results are also aligned to the experimental results in previous approaches [3, 1]. That is, our draft-and-revise decoding can preserve the global contexts of the draft images in the revise phase, although we use a small value of $T_{\text{revise}}$ such as 2 or 4. Note that the fine-grained details can be changed after infilling masks, since our method randomly samples the codes for unmasking based on the predicted softmax distribution over codes in Eq. 9.

### D.5 Societal Impact

Since our framework can learn large-scale datasets for a real-world applications, the model has a potential of negative social impacts by generating socially biased or violent images according to the training data. Thus, exhaustive filtering of training data and evaluation after training have to be conducted for avoiding social problems.