# OpenReview forum: "Draft-and-Revise: Effective Image Generation with Contextual RQ-Transformer"
_NeurIPS.cc/2022/Conference — NeurIPS 2022 Accept_

### Official Review · Reviewer_bucK · 2022-07-10

**Rating:** 4
**Confidence:** 4
**Soundness:** 3 good
**Presentation:** 4 excellent
**Contribution:** 2 fair

**Summary:**

This paper pointed out the limitation of the autoregressive image generation using the existing transformer based on the vector quantization methods and proposed the contextual RQ-transformer is trained to infer the masked code stacks by bidirectional self-attention like BERT in order to solve it. Furthermore, they suggested the two-phase decoding, Draft to fill the masked codes and Revise to increase the image quality, in order to break the trade-off between quality and diversity. They demonstrated their significant performances in class-conditional/text-conditional image generation as well as class-conditional image inpainting experiments.

**Questions:**

Overall, the proposed method is good, and the performance has also been well proven through many experiments. However, it is questionable that the novelty, which is the most important, is not enough and does not explicitly specify the differentiation from MaskGIT, which is the most like the problem raised. Being able to clearly explain the weakness mentioned above will be the biggest key to the final rating.

**Limitations:**

Please see the weaknesses and questions.

**Strengths And Weaknesses:**

### Strengths
- Overall clear presentation and discussion on the key argument/idea.
- The justifications and motivations are clearly established. Especially, the draft-and revise process that imitates the image generation process of a human expert is interesting.
- Overall, both qualitative and quantitative performance improvements over existing methods are convincing in various experiments.

### Weaknesses
- Novelty: It seems only to the extent of introducing the limitation and solution proposed by MaskGIT[6] to RQ-Transformer. Especially, for introducing the bidirectional transformer to the VQ scheme, the most novel part is the mask generation algorithm; MaskGIT first proposed this using VQVAE. However, this paper simply applies it to RQ-VAE in the draft phase and does not present any special novelty.
 For the above reasons, the comparison with MaskGIT is indispensable to verify their novelty. However, they hardly mentioned MaskGIT in both Related work and Experiment sessions, while highlighting only the merits in comparison with the autoregressive method (i.e., VQGAN) and the Diffusion methods.
  > In Tabel 1, MaskGIT shows almost similar scores compared to Contextural RQ-Transformer, although it uses fewer parameters. However, there is a lack of analysis with MaskGIT.

  > They highlight their inference speed compared with VQGAN at the 'trade-off between quality and sampling speed' sub-session. However, as far as I know, MaskGIT requires less iteration compared to Contextural RQ-Transformer in creation (Maskgit needs 16 iterations for a generation while Contextural RQ-Transformer needs T_draft = 64, T_revise = 2, M = 2.)

 I agree that the improvement of performance through several experiments is convincing. However, I believe that in-depth comparison and analysis in terms of motivation or performance with MaskGIT are essential to demonstrate their novelty.

- Revise phase: Another novelty of the paper is the Revise phase. However, it lacks a clear rationale for why the performance improves because they only perform a simple random uniform sampling. In the Revise phase, what percentage of the code to revise is an important factor (i.e., the number of codes for re-masking), but there is no mention of it. Please provide more details about the Revise phase to improve the novelty.

---

> ### Author Response · Authors · 2022-08-02
> **Authors' Response to Reviewer bucK**
>
> **[Novelty]**
> First of all, we agree with the reviewer’s comment about the necessity of the more detailed discussion about MaskGIT. We will sufficiently discuss the similarity and difference between our framework and MaskGIT in both Related Work and Experiments.
>
> Note that while we use the training objective of MaskGIT for RQ-Transformer, our main novelty lies in the two-phased decoding strategy, Draft-and-Revise, that is the key to the state-of-the-art performance on image generation. When we have simply applied the confidence-based decoding based on MaskGIT, the performance improvement after the revise phase is rather marginal due to the low diversity of generated images as shown in Appendix C. Instead of simply applying the confidence-based decoding, we carefully design our two-phased decoding to promote the diversity of generated images in the draft phase and improve the fidelity of images throughout the revise phase. Thus, our framework sets new state of the art, achieving both high fidelity and high diversity in image generation.
>
>
>
> **[Performance Comparison with MaskGIT]**
> In Appendix C, we compare our decoding method with confidence-based decoding methods (Top-C, Top-C-50%) of MaskGIT.
> Unfortunately, we could not reproduce the reported results of MaskGIT due to the lack of information about the training details and temperature annealing in the sampling process, where they are the key factors to reproduce the reported results. Thus, we simulate the decoding of MaskGIT using the random sampling based on confidence scores (Top-C-50%). In Table 3 of Appendix C, Top-C and Top-C-50% shows better precision than the random decoding. Although Draft-and-Revise decoding also improves the performance of Top-C and Top-C-50% after our revise phase, the performance improvements are marginal due to the low diversity of generated images. The low diversity of Top-C and Top-C-50% results from confidence-based decoding that often generates too simplified images, as shown in Figure 2 of Appendix C. In contrast, our random sampling generates high diversity of draft images, and their precision is significantly improved in the revised phase. Eventually, our two-phased Draft-and-Revise decoding generates high-quality images with both high fidelity and diversity and achieves state-of-the-art results.
>
>
> **[Highlighting MaskGIT in both Related Work and Experiments]**
> As discussed above, we will attach more descriptions to highlight MaskGIT in Related Work and Experiments.
>
>
> **[Trade-off between quality and sampling speed]**
> Our method can control the trade-off between the quality and sampling speed, while having from FID=5.41 with $T_\text{draft}=8$ to FID=3.45 with $T_\text{draft}=64$. Contrastively, although MaskGIT uses 16 iterations to achieve its best FID=7.51, MaskGIT cannot control the trade-off between the quality and sampling speed, as described in the original paper [6]. Compared with MaskGIT, we note that the controllability of the trade-off between the quality and sampling speed is our practical advantage for deploying a generative model for real-world, time-critical applications.
>
>
>
>
> **[Simple Random Uniform Sampling in the Revise Phase]**
> We use the simple random sampling to avoid the discrepancy between training and inference rules of the trained model, since the model is trained to infill masks with uniformly random patterns as in Eq. (11). As mentioned in Line 316-319, we suggest that a more sophisticated scheme for the revise phase is worth exploring in future work.
>
>
>
> **[Percentage of the Revised Code]**
> We have conducted an ablation study to evaluate the effect of the percentage of the code at each revise step as shown in Figure 4(b-c) and Line 264-279.

---

### Official Review · Reviewer_uiaY · 2022-07-11

**Rating:** 6
**Confidence:** 4
**Soundness:** 3 good
**Presentation:** 3 good
**Contribution:** 3 good

**Summary:**

This paper presents a Draft-and-Revise framework with a novel contextual RQ-Transformer module for the task of image generation. I think the proposed model is an extension of the previous method "RQ-VAE + RQ-Transformer" (Lee et al., Autoregressive Image Generation using Residual Quantization, CVPR 2022). Specifically, the newly proposed model changes the paradigm of AR model that gradually generate an image by attending to only precedent codes generated. It introduces random masking and gradually generate an image by attending to contextual codes. In addition, the original Spatial Transformer is replaced by Bidirectional Spatial Transformer and a draft-and-revise strategy is introduced to further improve the performance.

**Questions:**

See weaknesses above.

**Limitations:**

The authors have adequately addressed the limitations and potential negative societal impact of their work.

**Strengths And Weaknesses:**

Strengths:

+ The idea is interesting, it draws from CMLMs (Ghazvininejad et al., Mask-predict: Parallel decoding of conditional masked language models, EMNLP-IJCNLP, 2019) and skillfully combines with previous work to form an effective image generation model.
+ The experiments verify the effectiveness of the proposed method in both class-conditional and text-conditional image generation tasks.
+ The presentation is clear and well organized.

Weaknesses:

+ The effectiveness of the proposed model seems to largely depend on the existing model "RQ-VAE + RQ-Transformer" (Lee et al., Autoregressive Image Generation using Residual Quantization, CVPR 2022), the experimental results also seem to illustrate this point.
+ In addition, the technical contribution of this method seems to be insufficient, mainly using the technical route of the previous method.

---

> ### Author Response · Authors · 2022-08-02
> **Authors' Response to Reviewer uiaY**
>
> Thanks for the constructive comments to improve our study.
>
> Our technical contributions are not limited to adopting the BERT-style training objective to RQ-Transformer, but includes a new concept of Draft-and-Revise as the key to achieve state-of-the-art performance.  Different from the previous confidence-based decoding, we design the two-phased decoding that intends to increase the diversity of generated images and then improve their fidelity in the revise phase. Note that simply applying the confidence-based decoding of MaskGIT leads to a limited performance gain due to the lack of diversity after the revise phase in Appendix C. Contrastively, our Draft-and-Revise enables Contextual RQ-Transformer to fully exploit the global contexts in images, while generating the images with both high diversity and high fidelity. Consequently, Contextual RQ-Transformer significantly outperforms the previous RQ-Transformer despite using less number of parameters as in Table 1. Meanwhile, we also remark that our Draft-and-Revise, which is our key contribution for high performance, is not a tailored method to RQ-VAE, but compatible with VQ-VAE or VQGAN as shown in Appendix B.  Therefore, our Draft-and-Revise decoding lays foundation for future study, and we believe that it can be widely adopted to bidirectional transformers for various generative tasks.

---

### Official Review · Reviewer_EFXa · 2022-07-11

**Rating:** 5
**Confidence:** 4
**Soundness:** 3 good
**Presentation:** 3 good
**Contribution:** 3 good

**Summary:**

This work proposed a two-stage framework for image synthesis, which consists of RQ-VAE and contextual RQ-Transformer. The proposed framework is inherited from the prior work [23] but modifies the original auto-regressive pipeline with a bidirectional pipeline borrowed from BERT.

**Questions:**

Please address my concerns shown in the weaknesses.

**Ethics Review Area:**

["I don’t know"]

**Strengths And Weaknesses:**

Pros.
+ The results are good. Compared with RQ-VAE, the proposed method achieves better FID with smaller model sizes. But the results are not as good as GAN-based approaches such as styleGAN-XL.
+ The writing is easy to follow.

Cons.
- Lack of in-depth analysis and strong motivation for the proposed method. Both RQ-VAE [23] and the proposed method used an RQVAE as a tokenizer. The difference lies in the second-stage transformer that produces the RQ tokens. In [23], the tokens are generated in an auto-regressive way, while in this submission the tokens are generated via masked image modeling as BERT. By combining RQVAE and BERT, the proposed method is claimed to perform better than existing approaches. However, it is still unclear how the bidirectional masked token modeling outperformed autoregressive modeling. More ablation studies are preferred.
- Some details are missing. Implementation details on text-conditional image synthesis are missing.
- The paper reported class-conditional results on ImageNet. How about unconditional image synthesis performance?  It would be better to provide results on diverse datasets such as Imagenet and FFHQ. Also, the reported results are in a resolution of 256x256, and one concern is the quality in higher resolutions such as 1024x1024.

---

> ### Author Response · Authors · 2022-08-02
> **Authors' Response to Reviewer EFXa**
>
> **[Lack of In-Depth Analysis and Strong Motivation]**
> First of all, we summarize our two main contributions as follows.
>
> (1) We adopt the masked code stack modeling of RQ-VAE to Contextual RQ-Transformer for exploiting global contexts in images during generation.
>
> (2) We propose the novel two-phased decoding, Draft-and-Revise, to effectively control the quality-diversity trade-off in generated images and achieve state-of-the-art performance.  We intend to use a uniformly random partition in the draft phase to increase the recall of generated images instead of using confidence-based decoding. That is, masked image modeling alone does not warrant high performance, which is our key motivation.
>
> Compared with autoregressive (AR) models that are trained to use the only unidirectional (left-to-right) contexts during image generation, bidirectional transformers can exploit more contextual information and outperform previous AR models. In fact, Figure 1 shows that bidirectional transformers can exploit the global contexts during image generation, while the AR model cannot. In addition, our Draft-and-Revise decoding is the main contribution of our study for Contextual RQ-Transformer to fully exploit global contexts during image generation and outperform previous AR models as shown in our experiments in Table 1 and Appendix B, and ablation studies in Section 4.4 and Appendix C.
>
>
> **[Implementation Details of Text-Conditional Image Synthesis]**
> The implementation details of text-conditional image synthesis (Line 233-237) are described in Appendix A. We will add a sentence to refer readers to it.
>
>
> **[Unconditional Image Synthesis Performance]**
> We regret that it is infeasible to report the performance of unconditional image synthesis on ImageNet due to the time limit of the rebuttal period. Regarding the FFHQ dataset, we have evaluated preliminary experiments on FFHQ. Our Contextual RQ-Transformer (362M) has achieved marginally better FID=9.86 than RQ-Transformer [23] (370M) with FID=10.38, although Draft-and-Revise improves the qualitative results. This is mainly due to the fact that the FID score on the unimodal dataset such as FFHQ is unreliable, which is also shown in recent studies [NewRef-1, NewRef-2].
>
> [NewRef-1] Kynkäänniemi, T., et al. "The Role of ImageNet Classes in Fr\'echet Inception Distance." arXiv preprint arXiv:2203.06026 (2022).
> [NewRef-2] Morozov, S., et al. "On self-supervised image representations for GAN evaluation." International Conference on Learning Representations. 2020.
>
> **[Generating 1024x1024 Resolution of Images]**
> Since our model is not a tailored method for a specific resolution, our model can be trained with 1024x1024 images. However, training our model on 1024x1024 resolution of images is computationally demanding and infeasible to conduct the experiment during the rebuttal period. Meanwhile, one can use a super-resolution model to increase the resolution of generated images to 1024x1024, considering that recent large-scale text-to-image generation models such as DALL-E 2 [NewRef-3], Imagen [NewRef-4], and Parti [NewRef-5] also use super-resolution models for generating 1024x1024 images.
>
> [NewRef-3] Ramesh, Aditya, et al. "Hierarchical text-conditional image generation with clip latents." arXiv preprint arXiv:2204.06125 (2022).
> [NewRef-4] Yu, Jiahui, et al. "Scaling Autoregressive Models for Content-Rich Text-to-Image Generation." arXiv preprint arXiv:2206.10789 (2022).
> [NewRef-5] Saharia, Chitwan, et al. "Photorealistic Text-to-Image Diffusion Models with Deep Language Understanding." arXiv preprint arXiv:2205.11487 (2022).

---

### Official Review · Reviewer_7o7S · 2022-07-12

**Rating:** 7
**Confidence:** 3
**Soundness:** 3 good
**Presentation:** 3 good
**Contribution:** 4 excellent

**Summary:**

The paper achieves state-of-the-art generation quality on the ImageNet and CC-3M text2image synthesis task by proposing a new transformer architecture that models spatial attention and depth attention of Residual Quantization, and iteratively revising the generated image by randomly masking the output image and re-synthesizing the masked region.

**Questions:**

Why does [Draft only] achieve such bad FID? It looks like FID ranges from 50+ to 15 in Figure4(a). Shouldn't it still achieve similar FID as MaskGIT or RQ-Transformer? It would be nice if Contextual RQ-Transformer is also evaluated in Table 1 without the revise step. To me, Contextual RQ-Transformer sounds like the architecture described in the upper portion of Figure 2 without the revise step. So it is unclear if [Contextual RQ-Transformer] includes the revise step in Table 1.

**Limitations:**

The paper does not seem to discuss the societal impact of the paper.

**Strengths And Weaknesses:**

Strength.

* The paper achieves SOTA result on ImageNet synthesis, while making the inference time faster than previous methods including VQGAN.
* I find the idea of revising the generated image very interesting. The paper adds the notion of corse-to-fine refinement to transformer-based autoregressive models, in both the depth transformer of RQ and the revise step. There existed conceptual gap between AR and Diffusion-based models, in that AR models performed iterative generation spatially, and the diffusion-based models performed iterative generation in a coarse-to-fine manner. The proposed method developed a model that can control the iterative refinement in both axes independently, by changing T_draft and M.
* I like ablations and investigations including Figure 4.
* There is adequate explanation of previous literature, including Section A.1.

Weakness

* I found the exposition a bit difficult to follow. The paper introduces many new notations like T, D, M, and Pi. It would be nice if they are accompanied by textual description rather than just referring to them as symbols.
* The paper puts emphasis on parallel decoding of features, but in the end, the best performing model is using T_draft = 64, which doesn't use parallel decoding in the draft phase. This is confusing in some sense. Moreover, the paper could discuss more on its similarity to MaskGIT.

---

> ### Author Response · Authors · 2022-08-02
> **Authors' Response to Reviewer 7o7S**
>
> **[Textual Descriptions of Notations]**
> Thanks for the suggestion. We will add more textual descriptions for easy understanding of our paper.
>
> **[About Parallel Decoding in the Draft Phase]**
> Thanks for the constructive comment. In the revised version, we will explain about the role of parallel decoding in the draft phase to address your concern as follows.
>
> Parallel decoding is still important for time-critical applications, even if it can sacrifice the quality of generated images. Thus, to report the best performance of Contextual RQ-Transformer, we use $T_\text{draft}=64$ as  shown in Table 1. Note that the role of parallel decoding in the draft phase controls the trade-off between the sampling speed and the quality of images as shown in Table 4.  Specifically, the parallel decoding can accelerate the sampling speed by generating many code stacks at once, but the quality of generated images can be deteriorated due to the inconsistency of visual details between simultaneously generated spatial regions. Nevertheless, our Contextual RQ-Transformer with $T_\text{draft}=8$ achieves both faster sampling speed and better FID scores than previous methods.
>
>
> **[Bad FID of Draft only]**
> The reason why [Draft only] in Figure 4(a) has bad FIDs is that we **intend to** increase the diversity of generated images in the draft phase for maximizing the quality of generated images through the revise phase. We remark that our two-phase decoding, Draft-and-Revise, is designed to maximize the final performance after the revise phase, not the performance of [Draft only]. Thus, we do not apply any sampling strategy, such as temperature scaling, in the draft phase to increase the recall of generated images as shown in Figure 4(b). This leads to low precision scores and high FID scores. However,  our revise phase can significantly improve the precision in a few update steps (M) to achieve the best FID score. Meanwhile, if we use a confidence-based sampling strategy (Top-C-50%) for [Draft only] in Appendix C, [Draft only] has the competitive FID with MaskGIT and RQ-Transformer. However, its performance improvement after the revise phase is marginal due to the lack of diversity. Thus, we focus on increasing the diversity of draft images for the best performance after the revise phase.
>
>
> **[The Revise Step in Table 1]**
> Table 1 shows the performance of Contextual RQ-Transformer after the revise phase, which confirms that our two-phased decoding achieves the best performance after the revise phase.
>
>
> **[Societal Impact]**
> Since the proposed framework can learn large-scale datasets for a real-world application such as text-to-image generation, the model may generate a socially biased, violent, and sexual images according to the training data. Thus, exhaustive filtering of training data has to be conducted for avoiding potential social problems.

---

### Author Response · Authors · 2022-08-02
**We appreciate the reviewers’ constructive comments.**

We appreciate the reviewers’ constructive comments and sincerely respond to all the questions and concerns.
In addition, we have uploaded the revised version of our submission and highlighted the changes during the rebuttal period to address the reviews’ comments.
In the author's responses, the line numbers (e.g., Line 233-237) are based on the revised version of our paper.

---

### Meta-Review · Area_Chair_N6Hf · 2022-08-25

**Recommendation:** Accept
**Confidence:** Less certain

**Metareview:**

A new transformer method for image generation is discussed. Reviewers appreciated the results but raised concerns regarding exposition, some questionable ablations, limited novelty and relation to prior work (MaskGIT). The rebuttal was able to address some concerns. In a discussion reviewers generally kept their rating but raised concerns regarding novelty and ablations again. AC thinks the paper just barely made the cut and strongly encourages authors to further improve the ablations in the camera ready version to further strengthen the paper. AC also recommended senior ACs to look at this decision and possibly revise.

**Award:**

No

---

### Decision · Program_Chairs · 2022-09-14

Accept